# Classification of Pulmonary Damage Stages Caused by COVID-19 Disease from CT Scans via Transfer Learning

**DOI:** 10.3390/bioengineering10010006

**Published:** 2022-12-20

**Authors:** Irina Andra Tache, Dimitrios Glotsos, Silviu Marcel Stanciu

**Affiliations:** 1Automatic Control and Systems Engineering Department, Faculty of Automatic Control and Computers, University Politehnica of Bucharest, 060042 Bucharest, Romania; 2Department of Image Fusion and Analytics, Advanta, Siemens SRL, 15 Noiembrie Bvd, 500097 Brasov, Romania; 3Biomedical Engineering Department, Egaleo Park Campus, University of West Attica, 12243 Athens, Greece; 4Carol Davila University Central Emergency Military Hospital, 010825 Bucharest, Romania; 5Faculty of General Medicine, “Carol Davila” University of Medicine and Pharmacy, 8 Eroii Sanitari Bvd, 050474 Bucharest, Romania

**Keywords:** deep learning, transfer learning, convolutional neural network, medical imaging processing

## Abstract

The COVID-19 pandemic has produced social and economic changes that are still affecting our lives. The coronavirus is proinflammatory, it is replicating, and it is quickly spreading. The most affected organ is the lung, and the evolution of the disease can degenerate very rapidly from the early phase, also known as mild to moderate and even severe stages, where the percentage of recovered patients is very low. Therefore, a fast and automatic method to detect the disease stages for patients who underwent a computer tomography investigation can improve the clinical protocol. Transfer learning is used do tackle this issue, mainly by decreasing the computational time. The dataset is composed of images from public databases from 118 patients and new data from 55 patients collected during the COVID-19 spread in Romania in the spring of 2020. Even if the disease detection by the computerized tomography scans was studied using deep learning algorithms, to our knowledge, there are no studies related to the multiclass classification of the images into pulmonary damage stages. This could be helpful for physicians to automatically establish the disease severity and decide on the proper treatment for patients and any special surveillance, if needed. An evaluation study was completed by considering six different pre-trained CNNs. The results are encouraging, assuring an accuracy of around 87%. The clinical impact is still huge, even if the disease spread and severity are currently diminished.

## 1. Introduction

SARS-COV2 (severe acute respiratory syndrome coronavirus 2) in humans was first reported in Wuhan, China, on 31 December 2019; it is a single-strand ribonucleic acid virus from the Coronaviridae family [1]. It is a highly infectious disease, and its main symptomatology includes fever, dry cough, respiratory distress, and tiredness; it can worsen with difficulties in breathing, hence causing low oxygen saturation and chest pains. 

Three new, more severe coronaviruses have emerged from the animal reservoir in the last two decades: SARS-CoV in November 2002, MERS-CoV in September 2012, and SARS-CoV-2, which caused COVID-19 in China in December 2019 and was declared a pandemic by the WHO (World Health Organization) in March 2020 [2]. 

Scientists and medical equipment manufacturers, on the other hand, strive to use the latest innovative techniques to develop new systems based on artificial intelligence algorithms, modeling, and simulation. New funding sources are opening in the digitization of the health system, offering new opportunities for employment, training programs, and entrepreneurship.

The stages of lung damages caused by this disease are roughly split into mild, moderate, and severe. The moderate phase has similar symptoms as pneumonia, whereas the severe phase is difficult to distinguish from pulmonary cancer [3]. If a patient with mild lung symptoms is not treated correctly, the disease can quickly evolve to further stages and can even cause death. 

The standard medical protocol for patients’ detection is polymerase chain reaction (PCR) testing followed by scans, to monitor the progression of lung damage [4]. There are medical reports that suggest an axial computerized tomography (CT) scan is more reliable for the diagnosis of COVID-19 disease than X-ray radiography [5]. CT is a fast and frequently used medical procedure, especially in emergency cases, such as the evaluation of heart attacks [6]. 

As physicians are overwhelmed by the increasing number of cases every day all over the world, a fast, automatic detection of the disease stages from CT scans can help medical staff, which in many countries are outnumbered in comparison with the real demand. A computed-aided diagnosis will help with the overall management of the disease and save time and resources. Therefore, every step to increase and speed up the disease diagnosis has a huge social impact in these pandemic times. 

A faster classification of the disease stages will improve the efficiency of treatment administration. Depending on the lung damage stages, dedicated drugs must be administrated to a patient to prevent their symptomatology worsening. There is a high level of interest among scientists to develop algorithms and tools for helping medical staff in dealing with this pandemic disease. 

Artificial intelligence (AI) can improve the performances of the fast and automatic algorithms that tackle this problem [7]. The trade-off for medical AI applications is gathering data from multiple centers worldwide. Medical images, in a wide range of formats acquired from different types of medical equipment, were uploaded in renowned free databases to support the development of AI-based methods and to perform data classification or lung damage segmentation [8]. Moreover, researchers and medical centers worldwide are sharing data and knowledge via Kaggle competitions.

A couple of papers dealt with COVID-19 disease detection by X-ray radiography or CT scans. Considering X-ray radiography as the input dataset, in [9], a deep learning neural-network-based method, nCOVnet, was presented with the aim of detecting the lung damage caused by COVID-19 infection. Another strategy is using Shannon’s entropy and fractal theory to analyze image complexity for detecting the level of infection among different respiratory diseases, including COVID-19 [10]. In the Darwin AI open-source project [11], a dedicated convolution neural network (CNN) called CovidNet was developed and trained with normal, non-COVID-19, and COVID-19 patients via chest radiography. Finally, deep learning techniques are used on CT scans for the following purposes:-to classify images into normal and COVID-19: in [12], which uses for data augmentation the stationary wavelets; in [13], which uses a neural network, after automated selection of open lung images; and in [14] which presents nine different kinds of classification systems based on ML algorithms;-to differentiate the disease from common types of pneumonia: in [15], which uses pretrained networks such as VGG16 and RESNET50; in [16], which uses a multi-scale convolutional neural network with an area under the receiver operating characteristic curve (AUC) of 0.962; in [17], where normal cases were added, and Q-deformed entropy handcrafted features are classified using a long short-term memory network with a maximum accuracy of 99.68%;-to consider a data augmentation using a Fourier transform [18] for reducing the overfitting problems of lung segmentation. 

On the software market, Myrian COVID-19 is a protocol based on XP-Lung, an automatic lung segmentation tool of radiography and CT scans that aims to measure lung deficiency. The main characteristics of this software include the automatic calculation of global lung volume, visualization of healthy lung areas, visualization of pathological lung areas (opacity of ground glass, crazy pavement, consolidations, and emphysematous areas), automatic calculation of the lung reserve ratio, and complete lung volume density histogram. The tools were used in [19] to quantify the chest CT extent for cancer patients with a positive PCR test.

This paper presents a transfer learning approach for assessing lung damage stages on a new enhanced and labeled-from-scratch COVID-19 axial CT scans dataset. To the best of the authors’ knowledge, there are no papers dealing with lung damage stages’ classification caused by SARS-CoV2 virus from CT scans. The clinical impact will be important because a CT scan is the principal medical imaging for diagnosing the disease, so it will speed up the diagnosis and treatment of the patient. 

Regarding the methodology, the image database is improved by the data collected from a hospital dedicated to COVID-19 patients and from five available free datasets, e.g., the COVID-19 Open Research Dataset Challenge. Deep learning techniques have been used, and their classification results are compared in the Section 3. 

## 2. Materials and Methods

### 2.1. Medical Context

Coronaviruses are a family of viruses that cause mild and moderate diseases of the respiratory tract. From the six types of coronaviruses identified in humans, only two could degenerate into moderate to severe respiratory symptoms and lead to high rates of mortality: Middle East respiratory syndrome (MERS) and severe acute respiratory syndrome (SARS). The mean period of incubation is around 5.2 days [20].

The standard protocol for COVID-19 diagnosis (Figure 1) is a PCR test, which takes samples of saliva and mucus. The complementary protocol includes a chest X-ray, due to rapid screening and offering of insights into risk level and severity of the lung damage. 

As reported in [21,22,23], the most frequent COVID-19 patterns found in CT scans, by percentage, are ground glass patches (93.3%), followed by subpleural linear abnormalities (53.3%), consolidation patches (23.3%), and air bronchograms (23.3%), along with bronchial wall thickening (16.7%), crazy paving patterns (13.3%), and discrete nodules surrounded by ground glass appearance (10%). 

In Figure 2 presents the changes in lung tissues at different stages of the disease. As the disease evolves, the tissues are invaded by small patches of fluid.

### 2.2. Data Acquisition

CT or computed axial tomography is a medical image technique that uses multiple X-ray detectors located in certain positions to give measurements at different angles and finally build up the tomographic image of the investigated body part. It offers the possibility of 3D reconstruction of organs from images acquired in axial, coronal, and sagittal planes. It is dedicated to investigating different categories of organs, such as lungs, vessels, heart, brain, etc.

The image is stored in bytes because each gray value is an array of binary numbers. For example, the medical images need around 10 to 12 bytes per pixel, which means 1024 or 4096 different gray levels. The numbers of rows and columns of an image are carefully chosen by a compromise between the image storage and the capture of as many of the anatomical details as possible. The image dataset characteristics are summarized in Table 1.

### 2.3. Deep Learning Principles

Artificial intelligence techniques currently have a huge impact, and they are intensively used in medical image processing with very good results [24]. This has encouraged scientists to collect medical images acquired by different equipment and the development of dedicated convolutional neural networks.

Supervised learning tries to replicate human learning, for example, by solving optimization problems. Deep learning (DL) is usually applied on a lot of labeled data [25]. Hence, the more input data an algorithm has, the better the performances are. In the medical field, the data are not easy to collect, due to patient confidentiality policies. Therefore, to increase the amount of data, some strategies can be implemented; one is data augmentation, which applies on the raw data different transformations such as translation, rotation, reflection, etc., and another one is building an artificial dataset that resembles the initial data. Depending on the validation strategy, the dataset is split into training (more than 50% of data), validating, and/or testing.

Transfer learning principle is to use the predefined CNNs trained on thousands of images stored in the ImageNet database. This model is used to extract the general features of the objects by preserving the weights of the network layers [26]. The cut-off strategy will replace the top layers for feature extraction or fine-tuning. Feature extraction replaces only the top classificational layer with a dedicated new one, suitable for the problem to be solved. Fine-tuning replaces at least a top convolutional layer, followed by a classification layer, and trains the new CNN with the new dataset. The target will be the computation of the new weights of the updated layers for detecting the specific features of the objects found in the images.

Data augmentation (image resizing, rotation, translation, and different nonlinear filtering [27]) will help with expanding the dataset and, hence, avoiding overfitting problems, which occur when the neural network tends to memorize the training data, rather than extracting the general features of the objects. This step includes image resizing operations, rotations, translations, reflection, etc.

For optimization algorithm, in the current study, the stochastic gradient descending with momentum was used in fine-tuning approach because it updates the network parameters such as weights and bias, to minimize the cost function of a single element. It computes its gradient at each iteration, which makes it more efficient than the classical algorithm that calculates the sum of all examples [28]. For feature extraction, Adam method was preferred. 

The challenges facing a machine learning researcher are selecting the proper layers configuration and setting up the hyperparameters of the algorithm such as minibatch (the size of the data subset), learning rate, momentum, L2 regularization, number of epochs, etc. 

One of the most important and tedious steps in supervised learning is data labeling. Medical imaging experts are not easily found, so a lot of time is spent on their training because it requires a lot of medical practice. Labeling suffers from human subjectivity, which implies a team of experts is need for the labeling for serious DL studies. The labeling errors’ curation is very important to assure the best results. When this is not possible, it is better to remove a suspicious dataset from the study.

The evaluation metrics were also developed over time, to best approach the dataset amount. This includes the confusion matrix, from where the accuracy, precision, F1 score, sensitivity, etc., can be computed.

### 2.4. Classification Algorithm Overview

Even if there many medical images from normal or COVID-19 patients have been uploaded, it is sometimes still difficult to find good annotated medical images databases with a good spatial resolution.

The mandatory requirement of AI techniques is a large amount of labeled data. Therefore, efforts have been made to find CT scans from different regions all over the world via Kaggle competitions to provide proper requests to train dedicated neural nets. 

The present paper applies supervised learning to the multiclass classification problem. 

The dataset comprises new data from 55 patients acquired from a clinic during the rapid spreading of the disease in the spring of 2020, along with data collected from available online databases from Kaggle [29,30,31,32,33], summarizing a total of 7300 images. 

The dataset was collected from different sources and was acquired in the png, jpeg, and tif (after conversion from DICOM files) formats, in different sizes (from 140 × 140 pixels to 512 × 512), as can be seen in Figure 3. The complete dataset has been manually revised by a medical imaging expert to remove the images that are not corresponding to the open lung phase (Figure 4). 

The proposed algorithm is divided into the following steps:

Selection of the open lung views in the final dataset;

Normalize and further augment the data for the training dataset;

Find the top learnable layer and classification layer of the pre-trained neural network;

If the learnable layer is a fully connected layer replace it with a new one having a weight learn-rate factor of 10 and bias learn-rate factor of 10;

Otherwise, if the learnable layer is a convolution 2D layer, replace it with a new one having a weight learn-rate factor of 10 and bias learn-rate factor of 10;

Replace the top classification layer with a new one with 4 classes.

The input size for every simulation was in fact the required input layer size for each CNN.

Image augmentation was completed by using random reflection in the left-right direction, which means that each image was reflected horizontally with 50% probability, zoom range of 0.1, and horizontal and vertical translation with pixel range [−50, 50]. The models training options are presented in Table 2. 

The external validation was completed by splitting the dataset with the ratio 80:20 for training and testing (never seen by the training process at all). For assuring that there is not any mixing of images of the same patient into these two groups and for a good balance regarding the number of images and patients, respectively, this splitting was completed manually by a medical expert, after careful attention. For internal validation, the training dataset was further randomly split into 80:20 for adjusting the network weights and in-training validation (used only to check the metrics of the model after each epoch) with a fixed random seed. This last step was repeated 5 times to perform the 5-fold cross validation.

In addition to well-known evaluation metrics, the Cohen’s kappa coefficient and Matthew’s correlation coefficient (MCC) are computed. The kappa score measures the degree of agreement between the true values and the predicted values: a value of 1 means perfect agreement, and 0 means chance agreement. A value less than 0 means a disagreement, so the classifier is incorrect.

MCC applied for binary classification identifies the ineffectiveness of the classifier in classifying, especially for the negative class samples, and it has high values if the classifier is doing well. However, F1 score is highly influenced by which class is labeled as positive. Still, F1 score is different from MCC in respect to its magnitude, as the minority class is usually labeled as negative.

## 3. Results

The algorithm was implemented in a copy of MATLAB^®^2022a with an academic license. Considering the speeding up, the running time of the computational environment for these tests was completed on a supported GPU device, an NVIDIA GeForce 1650 with 4 GB RAM.

The dataset distribution into classes for the training, validation, and testing is organized in Table 3.

The global evaluation metrics, when considering five-fold cross validation, re given in Table 4. Considering the best performance model for Resnet50, the mean evaluation metrics for each class for five-fold cross validation are presented in Table 5. 

The plots for the convergence of accuracy and loss for both training and validation are presented in Figure 5. For a clinical investigation of the results, the predictions on validation images are represented by colors, considering red as the object and blue for the background, as shown in Figure 6.

For comparison, model 2 from [16] was considered with the following properties: Resnet50 with a retrained convolution base, followed by these layers: Global MaxPooling, a fully connected layer with 512 neurons with RELU as its activation function, a batch normalization, a dropout of 0.5, a fully connected layer with 512 neurons with RELU as its activation function, a batch normalization, a dropout of 0.5, and a fully connected layer with four neurons with softmax as its activation function. The mini-batch size was 128, and the number of epochs was 350. The training loss was 0.8435, the validation loss was 1.250677227973938, the training accuracy was 68.79, and the validation accuracy was 53.4; and a very small test accuracy with the plots of convergence is shown in Figure 7.

## 4. Discussion

The main contributions of the presented paper are as follows:development of a new classification algorithm for the suggestive aspect of COVID-19 lung damage into mild, medium, or severe stages, for the axial view of computed tomography images using deep neural networks;enhancement of the five different online databases with new images collected from 55 patients;manual selection of the open lung phase images and their labeling into four classes.

The main challenge was the poor quality of the available online datasets. There were cases of sample duplication when the data were merged. Moreover, some images were labeled with different landmarks or text indications. An imagistic expert was needed to revise, to identify the images belonging to each patient, and to label them into four classes. 

Due to restrictions on acquiring new medical data and to speed up the computational time, transfer learning via fine-tuning was used for 7300 images belonging to 173 patients. An evaluation study was completed by considering six different pre-trained CNNs. The results were compared regarding the precision, accuracy, recall, etc. 

The best model was obtained with Resnet50, by considering 30 epochs and 32 mini-batch size. The model performance was evaluated considering its accuracy of more than 86%. The Cohen’s kappa coefficient of 0.65 approached 1, which means a good degree of agreement between the true values and the predicted values. The Matthews’ correlation coefficient of 0.76 means an effectiveness in the classification of the four classes. 

The best matching class was the normal one, which is explainable considering the lung aspect, while the worst case was the mild class, which may be due to the rather small lung damages.

The main pitfall of deep learning is collecting more and more data for improving the models. A bigger dataset will not always produce better results, due to a lack of attention or expertise regarding the quality of the input data [34]. 

Hence, one of the limitations of this study is the significance of the database of images used to instruct the network (both qualitatively and quantitatively). Our future work will include a more robust instruction of the AI tools, with the type of data distribution and a possible dimension reduction like in [35].

## 5. Conclusions

Discovering ways to enhance the process of diagnosing and treating COVID-19 patients is a worldwide emulation. 

The proposed technical solution will impact our society in the battle against COVID-19 disease, in terms of the faster screening of patients. The medical protocol for diagnosing patients is a physical exam and PCR testing. Since the result of the PCR test can take almost 24 h, physicians can choose to examine the patients’ lung damage using CT scans. Having a powerful computing tool with an AI-enhanced diagnosis will assist medical staff, will improve clinical management, and will even improve treatment of COVID-19 disease. 

## Figures and Tables

**Figure 1 bioengineering-10-00006-f001:**
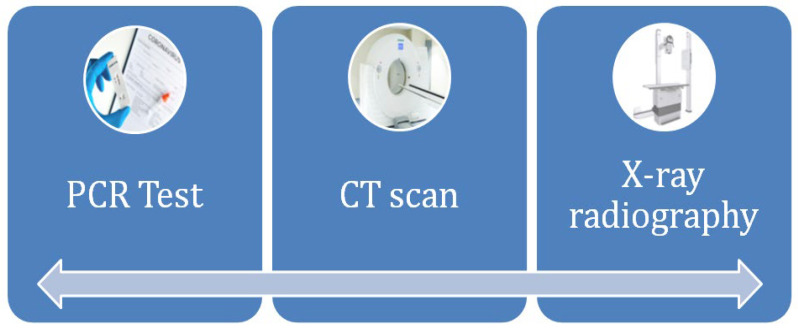
The standard protocol for COVID-19 diagnosis.

**Figure 2 bioengineering-10-00006-f002:**
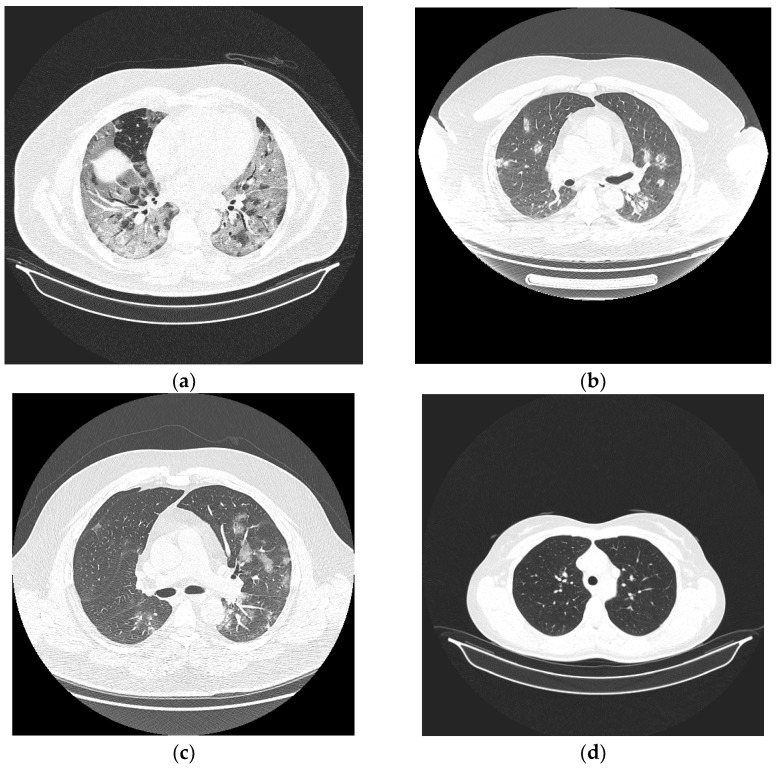
The evolution stages of COVID-19 infection in lungs. (**a**) Severe, (**b**) Moderate, (**c**) Mild, (**d**) Normal.

**Figure 3 bioengineering-10-00006-f003:**
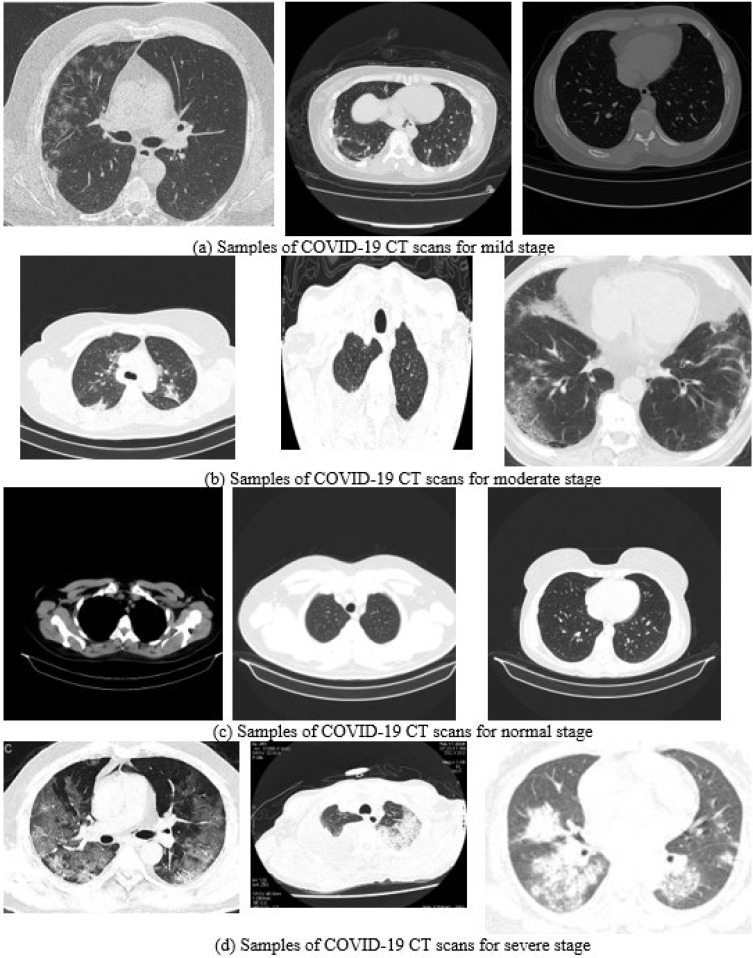
The diversity of the dataset.

**Figure 4 bioengineering-10-00006-f004:**
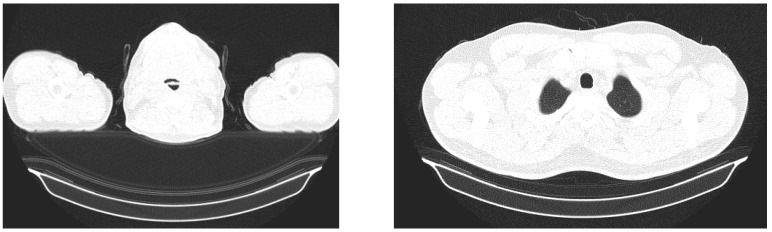
Samples of frames manually rejected from the dataset.

**Figure 5 bioengineering-10-00006-f005:**
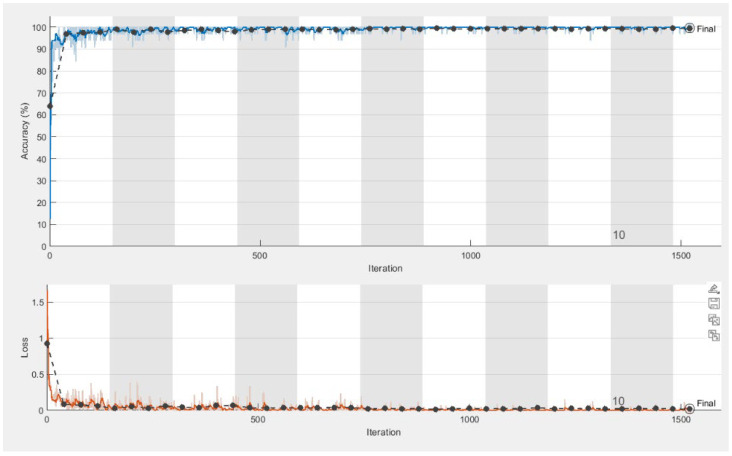
The plots for convergence of accuracy (**upper** graphic) and loss-(**bottom** graphic) for both training (in light blue for accuracy and in orange for loss) and validation (dotted).

**Figure 6 bioengineering-10-00006-f006:**
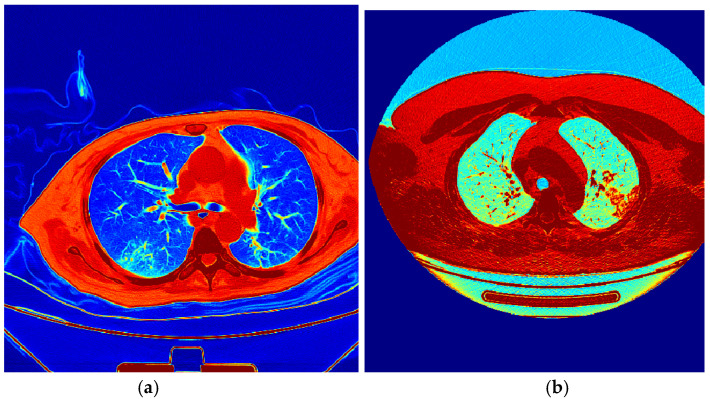
The classification results for (**a**) severe class and (**b**) moderate class meaning that red it’s a very important region and blue less important one used by the classifier.

**Figure 7 bioengineering-10-00006-f007:**
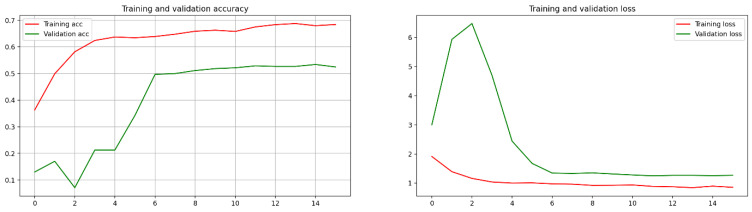
The plots of convergence for the comparison model.

**Table 1 bioengineering-10-00006-t001:** Parameters of CT scan acquisition.

Parameters	Unit	Values
Exposure Time	ms	600
Tube Current	mA	106
Color Type	-	grayscale
Bit Depth	-	12
Intensifier Size	mm	250
Image Width	pixels	512
Image Height	pixels	512
Number of frames per series	frames	200–350

**Table 2 bioengineering-10-00006-t002:** Training options.

No.	Hyperparameter	Value
1	Algorithm Type	Adam
2	No. of Epochs	30
3	Learn-Rate Schedule	piecewise
4	Learn-Rate Drop Factor	0.2
5	Learn-Rate Drop Period	5
6	Mini-Batch Size	32
7	Initial Learn Rate	10^−4^
8	Validation Frequency	40
9	Validation Patience	15
10	Shuffle	Every epoch

**Table 3 bioengineering-10-00006-t003:** The dataset distribution into classes.

No. Patients/No. Images	Total Images	Mild	Moderate	Normal	Severe
Training	4754	832	1021	2454	447
Validation	1188	208	255	613	112
Testing	1358	244	286	714	114

**Table 4 bioengineering-10-00006-t004:** The global evaluation metrics when considering 5-fold cross validation (mean value, min value, and max value).

Training Network	Accuracy (%)	Recall	Specificity	Precision	False Positive Rate	F1 Score	Matthews’ Correlation Coefficient	Cohen’s Kappa Coefficient
Resnet 50	86.89 (86.01,87.19)	0.8 (0.79,0.82)	0.96 (0.96,0.96)	0.81 (0.78,0.81)	0.04 (0.04,0.04)	0.79 (0.78,0.8)	0.76 (0.74,0.77)	0.65 (0.63,0.66)
Inceptionv3	85.99 (82.11,86.97)	0.78 (0.72,0.8)	0.96 (0.95,0.96)	0.8 (0.75,0.82)	0.04 (0.04,0.05)	0.77 (0.71,0.8)	0.74 (0.68,0.77)	0.63 (0.52,0.65)
Googlenet	83.4 (82.4,84.54)	0.76 (0.74,0.77)	0.95 (0.95,0.95)	0.78 (0.77,0.79)	0.05 (0.05,0.05)	0.74 (0.72,0.76)	0.71 (0.69,0.73)	0.56 (0.53,0.59)
Mobilenetv2	84.15 (83.65,86.75)	0.76 (0.74,0.81)	0.95 (0.95,0.96)	0.77 (0.76,0.8)	0.05 (0.04,0.05)	0.75 (0.74,0.8)	0.71 (0.7,0.76)	0.58 (0.56,0.65)
Squeenet	85.71 (84.17,87.78)	0.79 (0.76,0.81)	0.96 (0.95,0.96)	0.79 (0.77,0.82)	0.04 (0.04,0.05)	0.78 (0.75,0.81)	0.75 (0.72,0.78)	0.62 (0.58,0.67)
Shufflenet	80.65 (80.04,80.65)	0.73 (0.71,0.73)	0.94 (0.94,0.94)	0.73 (0.72,0.73)	0.06 (0.06,0.06)	0.69 (0.68,0.69)	0.66 (0.65,0.66)	0.48 (0.47,0.48)

**Table 5 bioengineering-10-00006-t005:** The mean evaluation metrics for each class when considering 5-fold cross validation for the best model (Resnet 50).

Class	Precision	Recall	F1 Score	False Positive Rate	Specificity
Mild (%)	67.816	92.376	78.07	9.712	90.288
Moderate (%)	78.214	54.334	63.738	4.16	95.84
Normal (%)	99.972	100	99.986	0.032	99.968
Severe (%)	77.568	74.738	76.018	2.01	97.99

## Data Availability

The data has been acquired as part of the projects acknowledged in the manuscript, and cannot be made public, considering GDPR regulations and the content of the informed consent signed by the patients.

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
