# Peer review of "Classification of Pulmonary Damage Stages Caused by COVID-19 Disease from CT Scans via Transfer Learning"

_bioengineering, 2022, doi:10.3390/bioengineering10010006_

Round 1
Reviewer 1 Report
This is an article that makes use of Transfer Learning for the classification of TC in COVID-19. I have to reject the article for the following reasons:
- Explain what many acronyms such as CNN and CT are. They appear in the text many times but can be many things.
- The definition of Deep Learning is not at all the right one. I suggest referring to the classic literature base for the problem.
- "The holdout method was the appropriate cross-validation approach for the current 219 large dataset." Hold-Out and Cross-Validation are different things.
- At some points they talk about thousands of images ("The total number of training images are 4995 and the ones of validation are 2203.") but at other points they say "219 large dataset". It is not clear how many elements this is.
- The text says "The data is randomly shuffled before dividing it into training and validation sets". No test set?
- If we take the number of elements per class from Table 4 as a reference. This is too little data for a model to generalize the information. Therefore, the evaluation strategy is not sufficient.
- It is not explained what is being done in Fine Tuning. Is the entire network being re-tuned? The Transfer Learning principle may not be fulfilled.
- A comparison of the models is made but not a statistical study of which one is better. It is not identified in the conclusions either.
- In general, I believe that the experimental part is scarce and that the domain knowledge regarding Artificial Intelligence is not demonstrated.
Author Response
I have answered each question separately and uploaded a new version with new results in the Results section and in red the modified text.
- Explain what many acronyms such as CNN and CT are. They appear in the text many times but can be many things.
I have modify especially in the discussion section were indeed it was confusing.
- The definition of Deep Learning is not at all the right one. I suggest referring to the classic literature base for the problem.
It was updated.
- "The holdout method was the appropriate cross-validation approach for the current 219 large dataset." Hold-Out and Cross-Validation are different things.
It was replaced by For model performance evaluation the holdout method was used as long as the external validation for the best selected models. The dataset splitting ratio was 70:30 on one hand or 80:20:20 on the other hand for training, validation and testing assuring that there are not mixing of images of the same patient into the these groups.
- At some points they talk about thousands of images ("The total number of training images are 4995 and the ones of validation are 2203.") but at other points they say "219 large dataset". It is not clear how many elements this is.
In the second time it was referring to the no of patients.
- The text says "The data is randomly shuffled before dividing it into training and validation sets". No test set?
I added the external validation for the best selected models.
- If we take the number of elements per class from Table 4 as a reference. This is too little data for a model to generalize the information. Therefore, the evaluation strategy is not sufficient.
The table was updated.
- It is not explained what is being done in Fine Tuning. Is the entire network being re-tuned? The Transfer Learning principle may not be fulfilled.
No, only the top conv layer.
- A comparison of the models is made but not a statistical study of which one is better. It is not identified in the conclusions either.
The comparison of models was based on their prediction accuracy. We are open to suggestions for other statistical approaches that the reviewer believes that could be used to compare the different models.
- In general, I believe that the experimental part is scarce and that the domain knowledge regarding Artificial Intelligence is not demonstrated.
The experimental part was improved with new tests.
Reviewer 2 Report
Authors have presented an interesting solution to detect affected regions due to COVID -19
1. Though the works sounds interesting, it is essential to highlight to the novelty of the proposed work.
2. The following work also does segmentation of affected regions due to COVID-19. Authors need to clearly specify how the pretrained based proposed model is better than the custom cnn proposed in the following works.
- A teacher–student framework with Fourier Transform augmentation for COVID-19 infection segmentation in CT images
Contour-enhanced attention CNN for CT-based COVID-19 segmentation.]
3. Statistics regarding dataset can be given as a table
4. The gaps in the existing works need to be consolidated and discussed at the end of related works section.
5. Research contributions need to be given at the end of the related works section.
6. Plots for convergence for loss and accuracy need to be presented.
Author Response
Dear reviewer,
I have answered each question separately and uploaded a new version with new results in the Results section and in red the modified text.
- The following work also does segmentation of affected regions due to COVID-19. Authors need to clearly specify how the pretrained based proposed model is better than the custom cnn proposed in the following works.
Thank you very much for the references, I have added them in the bibliography, but they are not useful for my assumed work. I only did a classification of the images, not semantic segmentation. My data was not labeled for segmentation tasks.
- Statistics regarding dataset can be given as a table
I have added a new table for this.
- The gaps in the existing works need to be consolidated and discussed at the end of related works section.
I have added new paragraphs in discussion section.
Research contributions need to be given at the end of the related works section.
I have stress them in discussion section
- Plots for convergence for loss and accuracy need to be presented.
Updated the figures.
Reviewer 3 Report
The paper is a lightweight study on the use of images of lungs affected by COVID to instruct neural networks to recognize the disease.
The paper ha some deficiencies and some potential for publication.
As to the deficiencies the first is that this idea is not that new and is already present in the literature. Not only but the authors do not provide enough evidence that this method can be better or faster in recognizing the disease with respect to alternatives.
All this said, I think that the less robust part of the study is about the significance of the database of images used to instruct the network (both qualitatively and quantitatively). The authors should at least discuss this point admitting that this is a limitation of their approach thus recognizing that more robust instruction of AI tools should entail the use of strategie like for example those cited in the two following papers (that I suggest to mention in this submission).
AA.VV. (2019). Is bigger always better? A controversial journey to the center of machine learning design, with uses and misuses of big data for predicting water meter failures.". J. Big Data 6 (1), 70. doi:10.1186/s40537-019-0235-y
AA.VV.. (2021). An alternative approach to dimension reduction for pareto distributed data: A case study. J. Big Data 8 (1), 39. doi:10.1186/s40537-021-00428-8
I suggest major revision.
Author Response
I have answered each question separately and uploaded a new version with new results in the Results section and in red the modified text.
As to the deficiencies the first is that this idea is not that new and is already present in the literature. Not only but the authors do not provide enough evidence that this method can be better or faster in recognizing the disease with respect to alternatives.
Our method is different than others in the sense that it tests many different models.
All this said, I think that the less robust part of the study is about the significance of the database of images used to instruct the network (both qualitatively and quantitatively). The authors should at least discuss this point admitting that this is a limitation of their approach thus recognizing that more robust instruction of AI tools should entail the use of strategie like for example those cited in the two following papers (that I suggest to mention in this submission).
Thank you very much for this comment, I have added these as limitations of my method and included the given references in the discussion section.
Round 2
Reviewer 1 Report
This is an article about the classification of the damage states of COVID-19 using Transfer Learning. I think the following issues should be taken into account:
- The text needs to be revised: "replicat-ing".
- Acronyms should be explained in the abstract.
- Is Fine Tuning done? What is fine tuned? The whole network?
- Why a 70-30 ratio for Hold Out? Is random splitting done? If so, how many times is this repeated to know at which values the results converge? If not random, how is the data split?
- Having multiple images of the same patient implies that there is dependent data, so it should be possible to know how the model is learning. I suggest pulling out more metrics like Cohen's Kappa to know what is going on.
- When comparing multiple models, there should be a statistical analysis of the results obtained.
Author Response
- The text needs to be revised: "replicat-ing".
Reply: Corrected
- Acronyms should be explained in the abstract.
Reply: Corrected
- Is Fine Tuning done? What is fine tuned? The whole network?
Reply: Yes. In the current study fine-tuning was applied on the whole network after replacement of at least a top learnable layer followed by a classification layer that will learn from the new dataset. The target will be the computation of the new weights of the updated layers for detecting the specific features of the objects.
Why a 70-30 ratio for Hold Out? Is random splitting done? If so, how many times is this repeated to know at which values the results converge? If not random, how is the data split?
Response
The external validation was done by splitting the dataset with the ratio 80:20 for training and testing (never seen by the training process at all). For assuring that there are not mixing of images of the same patient into these two groups and for a good balance regarding the number of images and patients respectively, this splitting was done manually by the medical expert, after a careful attention. For internal validation, the training dataset was further randomly split into 80:20 for adjusting the network weights and in-training validation (used only to check the metrics of the model after each epoch) with a fixed random seed. This last step was repeated 5 times for performing the 5-fold cross validation.
- Having multiple images of the same patient implies that there is dependent data, so it should be possible to know how the model is learning. I suggest pulling out more metrics like Cohen's Kappa to know what is going on.
Done, in the table 5.
- When comparing multiple models, there should be a statistical analysis of the results obtained.
Done, in the table 5 the average, min and max values are given.

Reviewer 2 Report
Results section need to be organized. Do not give implementation specific details in the paper directly. Instead convey those details systematically.
Author Response
Thank you, I have reorganised the results section

Reviewer 3 Report
I approve the paper
Author Response
Thank you
